# Genetic Association Study of Brain-Derived Neurotrophic Factor (BDNF) and Serotonin-Related Gene Variants in Suicide Attempters with Major Depressive Disorder and Control Persons

**DOI:** 10.3390/genes16111304

**Published:** 2025-11-01

**Authors:** Taehun Kim, Seong Beom Oh, Han Joo Choi, Chan Young Koh, Yong Oh Kim, Min Jeong Kim, Sijun Woo, Youngil Lee, Chang Min Lee, Jong Wan Kim, You-Shin Yi, Myung Ho Lim

**Affiliations:** 1Department of Emergency, Graduate School, Dankook University, Cheonan 31116, Republic of Korea; 2Department of Anatomy, College of Medicine, Dankook University, Cheonan 31116, Republic of Korea; 3Department of Neurology, College of Medicine, Dankook University, Cheonan 31116, Republic of Korea; 4Department of Laboratory Medicine, College of Medicine, Dankook University, Cheonan 31116, Republic of Korea; 5Department of Psychology and Psychotheray, College of Health Science, Dankook University, 119 Dandae-ro, Dongnam-gu, Cheonan 31116, Republic of Korea

**Keywords:** suicide, suicide attempt, major depressive disorder, *BDNF SLC6A4*, *TPH*, *HTR2A*

## Abstract

Objective: Suicide attempts have become one of the most serious problems worldwide in recent years. Suicide attempts are known to be genetically highly associated with serotonin and BDNF in international studies. Methods: In this study, we investigated the association with rs6265 (*BDNF*), and rs11030101 (*BDNF*), rs3813034 (*Serotonin Transporter*, *SLC6A4*) rs4680 (*TPH1*), rs1800532 (*TPH1*), rs1386493 (*TPH2)*, rs6311 (*HTR2A*), rs6313 (*HTR2A)*, rs6314 (*HTR2A*), rs6296 (*HTR1B*), gene polymorphism in Korean suicides with major depressive disorder who visited the emergency room. Results: A case-control study was conducted with 226 suicide and suicide attempts with major depressive disorder and 1469 general controls, and genotypes were determined by PCR-RFLP method. Significant associations between the rs11030101 (*BDNF*) genotype and suicide/suicide attempters were confirmed (OR 0.44, 95% CI 0.23–0.84, *p* = 0.0004), and significant associations between the rs11030101 (*BDNF)* allele and suicide/suicide attempters were confirmed (OR 0.63, 95% CI 0.50–0.80, *p* = 0.0002), in addition to the recessive genotype (OR 0.55, 95% CI 0.29–1.03, *p* = 0.03) and overdominant genotype (OR 0.66, 95% CI 0.49–0.88, *p* = 0.0046). Also, significant associations between the rs3813034 (*SLC6A4*) recessive genotype and suicide/suicide attempts were confirmed in the study results (OR 2.03, 95% CI 1.12–3.68, *p* = 0.028). After correction for multiple testing, none of the serotonin-related polymorphisms displayed any significant case–control differences. Conclusions: The *BDNF* gene may influence suicide attempts in Korean patients with major depression who have performed a suicide attempt. Replication in larger studies and other populations is needed.

## 1. Introduction

In 2021, there were 13,352 suicide deaths in Korea, making it the highest suicide rate (26.0 per 100,000 people) among countries in the Organization for Economic Co-operation and Development (OECD) [1]. Furthermore, while most OECD countries have seen a decline in suicide rates since 1990, South Korea has experienced an increase. While various biological, psychological, sociological, and economic factors contribute to the complex etiology of suicidal behavior, evidence for genetic factors is robust [2]. The most important determinant in those who attempt suicide is the individual’s genetic sensitivity to suicidal behavior. There was sufficient epidemiological evidence to show that suicidal tendencies are practiced within the family regardless of the presence of mental illness in family, twin, and adoption studies [2,3].

Over the past few decades, several studies have reported abnormalities in the function of the serotonin operating system associated with suicidal behavior [4]. Consequently, genes encoding proteins that regulate the neurotransmission of serotonin and *BDNF* (including Tryptophan hydroxylase *TPH1* and *TPH2*, serotonin transport gene *SLC6A4*, and serotonin receptor genes *HT1A*, *HTR2A*, and *HTR1B*) have been considered strong candidates in the association studies of suicidal behavior [4,5,6].

Studies, including postmortem and epigenetic expression studies, have reported that brain-derived *BDNF* is associated with suicidal behavior. A Slovenian research team reported an association between *BDNF* rs6265 polymorphism and suicide in completed suicides. Perroud et al. [7] investigated the interaction between *BDNF* rs6265 and a history of childhood sexual abuse. They found that violent suicide attempts were associated in adults with the Val/Val genotype of rs6265 when accompanied by a history of childhood sexual abuse. Sarchiapone et al. [8] reported an association between *BDNF* variants and suicide attempt risk in depressed patients when levels of childhood emotional, physical, and sexual abuse were high. More recently, Zouk et al. [9] found an association between the rs4923463 genotype allele GA and violent suicide attempts (*p* = 0.03) in patients with bipolar disorder. Zai et al. [10] conducted a meta-analysis of the Val66Met (rs6265) functional *BDNF* marker in suicide attempters across 12 studies (total N = 3352, of whom 1202 had a history of suicide). The results indicated that the Met allele and the Met-carrier genotype tended to increase the risk of suicide (*p* = 0.032, or Met = 1.16, 95% CI 1.01–1.32). Recently, Ratta-Apha et al. [11] conducted a meta-analysis including six studies conducted in Asia. The results showed a tendency for the Met allele to be associated with the risk of suicide attempts (number of Met alleles = 437: total number = 1428, pooled OR = 1.37, 95% CI = 1.01~1.86, Z value = 2.047, *p* = 0.041).

*5-HTTPR* (SLC6A4) has been extensively studied as a candidate gene for suicide vulnerability. Results from a recent meta-analysis demonstrated a strong association between the 5-*HTTPR* S allele and suicidal behavior [12]. A stronger difference was observed between suicide attempters and non-suicide attempters compared to healthy controls, indicating an association between the SNP variant and suicide independent of psychiatric diagnoses.

The *TPH1* gene is the first serotonin gene to be identified in association with suicide.

Early studies investigating variants within the *TPH1* gene reported that a polymorphic change in intron 7, a substitution from A to C at nucleotide 779 (A779C), was associated with violent suicide attempts [13]. Following this initial report, subsequent studies reported an association between the A779C variant and suicide attempts or completed suicides. Later research reported an increased frequency of the A779C allele in suicide attempters [14,15,16]. Rujescu et al. [17] and Bellivier et al. [18] found a significant association between the A218C single-nucleotide polymorphism (SNP) allele and suicide risk in Caucasians in meta-analyses of seven and nine studies, respectively. Li and He [19] confirmed a significant association between suicidal behavior and the A779C/A218C polymorphism in a review study including 22 prior studies, observing a stronger association in Caucasian samples. As the second isoform of *TPH*, *TPH2* plays a more dominant role in brain serotonin synthesis, making it a promising candidate gene in genetic studies of suicide [20]. In 2004, Zill et al. [21] examined associations between 263 suicide victims and healthy controls in Germany. They first identified associations between completed suicides and a single SNP (rs1386494) (*p* = 0.001) and a haplotype (*p* < 0.0001) [22]. In this study, they used 10 single SNPs to explore a 28 kb region of the *TPH2* gene where linkage disequilibrium (LD) increases and reported the presence of a haplotype block. Similarly, Lopez et al. [23] investigated four SNPs (rs11178997, rs1386494, rs1007023, and rs9325202) and reported associations between suicide attempts and these SNPs in a sample of 670 individuals from families with bipolar disorder.

Musil et al. [24] reported that the *TPH2* rs1386494 C/T polymorphism consistently showed a significant association with treatments for emergency suicide ideation (TESI) compared to controls (*p* = 0.0173). The *TPH2* rs1386494 C/T polymorphism showed a significant association with suicidal behavior in logistic regression analysis (*p* = 0.0041), with an odds ratio of 5.64 (95% CI 1.77–19.58).

The *HTR1A* gene, located on the long arm of chromosome 5, was proposed as a candidate gene for suicide susceptibility over the past decade. A study by Lemonde et al. [25] demonstrated an association between suicide attempters and the C(-1019) G (or rs6295) variant, where the GG genotype was four times more prevalent. These findings were replicated by a study showing increased methylation in the promoter region of the *HTR1A* gene in the prefrontal cortex of suicidally depressed patients carrying the C(-1019)G GG genotype [26]. Another study found the G allele occurred more frequently in suicide victims compared to controls and was associated with the number of stressful life events in the suicide victim group [27].

The *5-HT2A* receptor gene (HTR2A), located at 13q14q21, is known as an important candidate gene for suicide risk. The most extensively studied polymorphisms in *HTR2A* are A-1438G (rs6311 C/T) and T102C (rs6313). Li and He [19] concluded in a meta-analysis of 25 previous studies, including those examining *HTR2A* gene null associations, that there was no significant association between the T102C/A-1438G polymorphisms and suicidal behavior. Some studies observed a positive association between *HTR2A* markers and suicide. Giegling et al. [28] found that the *HTR2A* variants (rs594242-rs6311: G-C and rs6311C) were associated with increased nonviolent and impulsive suicidal behavior, respectively. Additionally, the CC genotype of the T102C SNP was more frequently found in suicide attempters.

Reviewing studies related to the *HTR1B* gene, the single variant rs11568817 and the haplotype variant rs130058 showed a significant association with suicide in a Chinese patient group with major depressive disorder [29].

Therefore, we selected 10 gene candidate polymorphisms related to serotonin based on previous studies related to these suicide attempts. The aim of this study was to investigate the association between the genetic type and alleles for the *TPH1*, *TPH2*, *SLC6A4*, *HTR1A*, *HTR2A*, and *BDNF* genes in Korean suicide attempters with major depressive disorder (suicide attempters) and a control group.

## 2. Methods

### 2.1. Study Population

Patients who had visited the emergency room of the Dankook University Hospital in connection with a suicide attempt were investigated within a week by emergency physicians and psychiatrists. These doctors conducted basic epidemiological questions and medical interviews to obtain age, gender, and method of suicide attempt, as well as diagnosis according to the *Diagnostic and Statistical Manual of Mental Disorders*, edition 5 (DSM-5) criteria. Suicide attempts were defined as self-harm behavior involving at least minimal suicidal intent. Patients diagnosed with intellectual disability, organic brain disorder, dementia, schizophrenia, or alcohol dependence were excluded. The final patient group comprised 226 patients with major depressive disorder. The control group (n = 1469) was recruited from the general population in ten South Korean cities among children and adolescents who reported no neuropsychiatric disorders. All participants, as well as the parents/legal guardians of the children, received a thorough explanation of the study’s purpose and provided written informed consent. The study was conducted with approval from the Institutional Review Board of Dankook University Hospital (2016-08-002). The demographic characteristics of the study subjects are presented in Table 1.

The genetic characteristics of polymorphisms are shown in Table 2.

### 2.2. Research Methods

#### 2.2.1. Epidemiological Questionnaire

This consists of basic survey questions about the subject, including gender, age, and method of suicide attempt. It was based on the emergency room chart records, where the attending physician questioned the subject.

#### 2.2.2. Genetic Testing

Blood samples were collected at the university hospital’s Department of Laboratory Medicine, and DNA was extracted from leukocytes using a commercial DNA extraction kit: the Wizard Genomic DNA Purification Kit (Promega, Madison, WI, USA).

Candidate genes selected included *TPH1/TPH2, SLC6A4, HTR2A*, and *BDNF*, specifically *TPH1* rs1800532 (*TPH1*), *TPH2* rs1386493, Serotonin Transporter (*SLC6A4*) rs3813034, *HTR2A* rs6311, rs6313, rs6314, *HTR1B* rs6296, and *BDNF* rs6265, rs11030101.

Gene testing was performed according to the procedures of our previous study as follows [30]: Immobilization of genomic DNA onto streptavidin-coated magnetic beads; assay oligonucleotide extension and ligation; PCR amplification; PCR product preparation; array hybridization and imaging were outsourced to Macrogen company^®^ (Seoul, Republic of Korea) [31]. This company delivered genotypes with quality scores calculated by proprietary Illumina^®^ algorithms [31].

#### 2.2.3. Data Analysis

Data were processed using SPSS 22.0 (Korean version). For statistical analysis, chi-square cross-tabulation was performed as needed for epidemiological surveys (e.g., gender) and to compare polymorphism frequencies between groups. Age comparisons with the control group used *t*-tests. *p*-values < 0.05 were considered statistically significant. The web-based statistical program SNPstats (http://www.snpstats.net/start.htm, accessed on 30 May 2025) was used to compare genotype and allele frequencies between the suicide attempt group and the control group, adjusted for age. Hardy–Weinberg equilibrium was verified for each analyzed result. Furthermore, Haploview software was used to analyze linkage disequilibrium relationships between genetic polymorphisms (Figure 1).

## 3. Results

Analysis of the association between *SLC6a4* rs3813034 and suicide attempters showed no significant association in genotype frequency (OR = 2.10, 95% CI = 1.15–3.85, *p* = 0.072), and a significant association was observed in the frequency of the recessive genotype among suicide attempters (OR = 2.03, 95% CI = 1.12–3.68, *p* = 0.03), but these significant values disappeared after the Bonferroni test (Table 3).

Analysis of the association between *BDNF rs11030101* and suicide attempters revealed a significant association in genotype frequency with suicide attempters (OR = 0.44, CI = 0.23–0.84, *p* = 0.0004), and the allele frequency also showed a significant association with suicide attempters (OR = 0.63, CI = 0.50–0.80, *p* = 0.0002). Furthermore, analysis of the association between the dominant genotype of *BDNF* rs1103010 and suicide attempters showed a significant association (OR = 0.57, CI = 0.43–0.76, *p* = 0.0004), and the analysis of the association between the heterozygous genotype and suicide attempters also showed a significant association (OR = 0.66, CI = 0.49–0.88, *p* = 0.0046). These results showed statistically significant values after the Bonferroni test (Table 3).

## 4. Discussion

Previous studies on the association between suicide attempters or completers and *BDNF* gene polymorphisms were primarily a number of studies on rs 6295 (*Val66Met*).

However, this study did not observe a significant association at rs6265. In previous studies in the past, a Slovenian study reported a significant association between rs6265 and suicide [32], and Zai et al. [10] also confirmed an association between the rs6265 allele and genotype and suicide risk in their meta-analysis. Furthermore, Ratta-Apha et al. [11] confirmed a significant association between the rs6265 allele and suicide perpetrators in six meta-studies conducted in Asia. However, Zarilli et al. [33] did not observe a significant association between the BDNF gene polymorphism rs6265 and suicide deaths. Furthermore, Sears et al. [34] did not observe a significant association between *BDNF* gene polymorphisms and suicide attempters with bipolar disorder. Similarly, a study in East Asian Chinese subjects did not find a significant association between *BDNF* gene polymorphisms and suicide attempters with unipolar or bipolar disorder. The findings of this study regarding the rs6265 gene polymorphism were inconsistent with those of Zai et al. [10] but consistent with those of Zarilli et al. [33].

In this study, a significant association was found between the frequency of the *BDNF rs11030101* genotype and suicide attempters, and a significant association was confirmed between the allele and suicide attempters. Furthermore, significant associations were also observed with the recessive genotype and the overdominant genotype; these results showed statistically significant values after the Bonferroni test. In previous studies on *BDNF rs11030101*, a significant association with asthma was first identified, followed by subsequent studies primarily confirming associations in psychiatric disorders. Pae et al. [35] reported that rs11030101 influences treatment responsiveness in a patient group comprising major depressive disorder, bipolar disorder, and schizophrenia. Tsai et al. [36] reported that rs11030101 affects the weight loss effect in antipsychotic drug response. Furthermore, Cao et al. [37] also reported that subjects with *BDNF* gene polymorphism variants decreased hippocampal volume, and Viikki et al. [38] reported that *BDNF rs11030101* influences the efficacy of electroconvulsive therapy in treatment-resistant depressive disorder. Additionally, Kwon et al. [30] reported an association of *BDNF rs11030101* in children with ADHD. Han et al. [39]. suggested that BDNF polymorphic variants may affect the Uncinate fasciculus region of the brain, which is a brain region associated with major depressive disorder. Also, it has also been reported that *BDNF* acts as a transducer between antidepressants and changes in neurons to improve symptoms of depression [40]. Since *BDNF* rs11030101 polymorphism belongs to the encoding region, it is estimated that it may affect depression or suicide attempts as an indirect association with rs6265 polymorphism, which is a close LD relationship, rather than a direct function. This study is the first to confirm an association between *BDNF rs11030101* and suicide attempters, both domestically and internationally.

In this study, a significant association was found between the frequency of the SLC6A4 rs3813034 recessive genotype and suicide attempters. However, when the Bonferroni correction for multiple testing is applied, these significance values disappeared. The serotonin transporter gene *SLC6A4* rs3813034 was initially found to be significantly associated with panic disorder [41] and showed a significant association with antidepressant treatment response in patients with major depressive disorder. Additionally, Enoch et al. [42] reported a significant association with suicidal behavior accompanied by childhood trauma. Meanwhile, *SLC6A4* rs3813034 also showed a significant association with personality traits characterized by sensitivity to pain, and suicidal behavior appears to be related to sensitivity to pain as well. This result is inconsistent with the international research by Enoch et al. [42].

This study had the following methodological limitations. First, this study has limitations in generalizing the findings as representative of Korea, as the patient group was smaller than the control group, and a significant age difference existed. Although a sufficient number of controls were secured, the genetic data were obtained prospectively from a cohort study conducted over a long period, making it impossible to match the suicide attempt group in terms of size and age. The younger age of the control group in our study still includes the potential risk of future suicide attempts, even if statistical adjustment was performed. Second, the patient group data was obtained from Cheonan and its surrounding areas, with a population of approximately 600,000, while the control group data was obtained from a broader sample across 10 cities nationwide, potentially introducing regional differences. South Korea is a single ethnic nation and has a relatively small territory, but there may still be cultural differences between regions. Considering that socioeconomic and sociocultural status is an epidemiological factor that influences suicide, it would also be necessary to confirm whether the patient group was evenly distributed across urban and rural areas. Third, while this study excluded individuals with severe neuropsychiatric disorders such as intellectual disability, dementia, schizophrenia, and alcohol dependence, it failed to assess epidemiological variables influencing suicide, such as family history, suicide attempt methods, characteristic traits, and trauma experiences, etc. Furthermore, due to difficulties in recruiting sufficient numbers of suicide attempters in actual clinical settings, this study focused on those with comorbid depressive disorders. Future studies should recruit a large group to compare the association between suicide attempters and genotypes or alleles within a single disease group. Meanwhile, Serin et al. [43] conducted psychiatric diagnoses on 48 suicide attempters, suicide completers, and controls, particularly in suicide attempters with high schizophrenia, paranoia, depressive disorder, and antisocial personality disorder scale scores. Therefore, suicide completers and attempters may have distinct characteristics. While this distinction is challenging in general clinical practice, it may be feasible in emergency room settings. Future studies should account for these epidemiological characteristics. Furthermore, this study did not examine biological markers such as serotonin-related neurotransmitters. Since the precise mechanism linking genetic polymorphisms themselves to suicide attempts or whether mental disorders like major depressive disorder mediate the effect on suicide attempts remains unclear, the conclusions of this study require further refinement.

## 5. Conclusions

Nevertheless, this study holds significance as it is the first international report to identify an association between *BDNF rs11030101* and suicide attempters. It also confirms an association between *SLC6A4* and suicide attempters, which is consistent with previous international studies.

## Figures and Tables

**Figure 1 genes-16-01304-f001:**
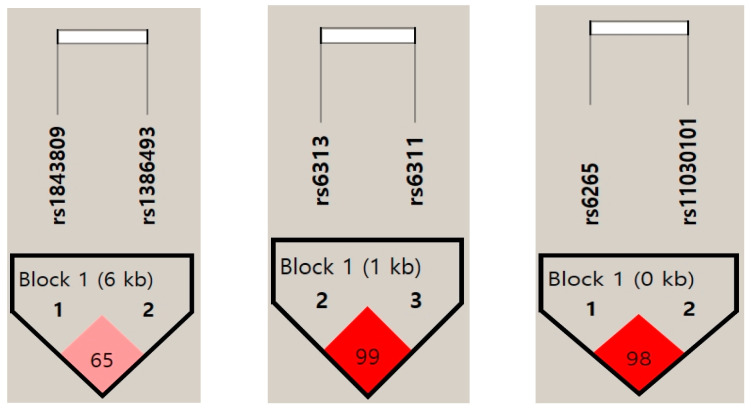
Linkage disequilibrium (LD) plot of single-nucleotide polymorphisms (SNPs) of TPH2, HTR2A, and BDNF. Numbers represent LD between SNP markers based on the D’ values. TPH2: Tryptophan hydroxylase 2 gene, HTR2A: Serotonin receptor 2A gene, BDNF: brain-derived neurotrophic factor gene.

**Table 1 genes-16-01304-t001:** Epidemiological characteristics between the suicide attempt group and the comparison group.

Rating Scale	Suicide Attempter Group (n = 180) Mean ± S.D.	Comparison Group (n = 159) Mean ± S.D.	t or x^2^	*p* Value
**Age ^a^**	43.44 ± 18.31	10.08 ± 1.23	73.08	<0.01
**Sex (N,%) ^b^**				
Female	111(49.11%)	801(54.53%)	2.31	0.13
Male	115(50.88%)	668(45.47%)		

These data represent mean ± S.D., by independent *t* test ^a^, or N (%), by chi-square test ^b^, significant *p* value < 0.05.

**Table 2 genes-16-01304-t002:** SNPs considered in this study.

SNP ID	Chromosome	Location	Position (Coordinate)	Distance	Alleles	
**rs6265**	11	UTR	27636491	128/93	G/A	BDNF
**rs11030101**	11	UTR	27637319	26/31	A/T	BDNF
**rs1386493**	12	Intron	70641445	−11,120	G/A	TPH2
**rs3813034**	17	flanking_3UTR	25548929	−102	C/A	SLC6A4
**rs6311**	13	flanking_5UTR	46369478	−1303	G/A	HTR2A
**rs6313**	13	Coding	46367940	310/101	C/T	HTR2A
**rs6314**	13	Coding	46307034	62/740	C/T	HTR2A
**rs6296**	6	Coding	78228978	312/860	G/C	HTR1B

BDNF: Brain-derived neurotropic factor; NCBI gene ID (Accession) is 2562 (NR002832 and NM170733).

**Table 3 genes-16-01304-t003:** Multivariate model for genotype distribution and allele frequencies in the suicide attempter group and comparison group.

Characteristics	Control		Suicide Attempter		OR	95% C.I.	*p*
	N	%	N	%			
**rs1800532** ** *TPH1* ** ** *Genotype* **							
AA	375	25.68	64	28.32	1.00		
AC	719	49.25	115	50.88	0.94	0.67–1.30	0.34
CC	366	25.07	47	20.8	0.75	0.50–1.13	
** *Allele* **							
A	1469	50.31	243	53.76			
C	1451	49.69	209	46.24	0.87	0.71–1.06	0.17
** *Dominant* **							
A/A	375	25.68	64	28.32	1.00		0.4
C/A, C/C	1085	74.32	162	71.68	0.87	0.64–1.20	
** *Recessive* **							
A/A, C/A	1094	74.93	179	79.2	1.00		0.16
C/C	366	25.07	47	20.8	0.78	0.56–1.11	
** *Overdominant* **							
A/A, C/C	741	50.75	111	49.12	1.00		0.65
C/A	719	49.25	115	50.88	1.07	0.81–1.41	
**rs1386493** ** *TPH2* ** ** *Genotype* **							
G/G	1093	74.86	172	76.44		1.00	
G/A	335	22.95	48	21.33	0.91	0.65–1.28	0.86
A/A	32	2.19	5	2.22	0.99	0.38–2.58	
** *Allele* **							
G	2521	86.34	392	87.11		1.00	
A	399	13.66	58	12.89	0.93	0.70–1.26	0.65
** *Dominant* **							
G/G	1093	74.86	172	76.44		1.00	
G/A, A/A	367	25.14	53	23.56	0.92	0.66–1.28	0.61
** *Recessive* **							
G/G, G/A	1428	97.81	220	97.78		1.00	
A/A	32	2.19	5	2.22	1.01	0.39–2.63	0.98
** *Overdominant* **							
G/G, A/A	1125	77.05	177	78.67		1.00	
G/A	335	22.95	48	21.33	0.91	0.65–1.28	0.59
**rs3813034** ** *SLC6A4* ** ** *Genotype* **							
C/C	954	65.88	139	61.33		1.00	
C/A	445	30.73	72	32	1.11	0.82–1.51	0.072
A/A	49	3.38	15	6.67	2.10	1.15–3.85	
** *Allele* **							
C	2353	81.25	350	77.33		1.00	
A	543	18.75	102	22.67	1.26	0.99–1.61	0.06
** *Dominant* **							
C/C	954	65.88	139	61.33		1.00	0.2
A/C, A/A	494	34.12	87	38.67	1.21	0.91–1.61	
** *Recessive* **							
C/C, A/C	1399	96.62	211	93.33		1.00	0.028
A/A	49	3.38	15	6.67	2.03	1.12–3.68	
** *Overdominant* **							
C/C, A/A	1003	69.27	154	68		1.00	0.73
A/C	445	30.73	72	32	1.05	0.78–1.42	
**rs6311** ** *HTR2A* ** ** *Genotype* **							
AA	392	26.8	52	23		1.00	
AG	714	48.8	112	49.6	1.18	0.83–1.68	0.39
GG	356	24.4	62	27.4	1.31	0.88–1.95	
** *Allele* **							
G	1426	51.23	216	47.56		1.00	
A	1498	48.77	236	52.44	1.04	0.85–1.27	0.70
** *Dominant* **							
A/A	392	26.81	52	23.11		1.00	
A/G, G/G	1070	73.19	174	76.89	1.23	0.88–1.71	0.22
** *Recessive* **							
A/A, A/G	1106	75.65	164	72		1.00	
G/G	356	24.35	62	28	1.17	0.86–1.61	0.32
** *Overdominant* **							
A/A, G/G	748	51.16	114	51.11		1.00	
A/G	714	48.84	112	48.89	1.03	0.78–1.36	0.84
**rs631364** ** *HTR2A* ** ** *Genotype* **							
T/T	361	24.59	53	23.56		1.00	
T/C	712	48.5	110	48	1.15	0.81–1.63	0.42
C/C	395	26.91	53	28.44	1.30	0.88–1.93	
** *Allele* **							
T	1502	51.16	216	47.56			
C	1434	48.84	236	52.44	1.14	0.94–1.40	0.18
** *Dominant* **							
T/T	395	26.91	53	23.56		1.00	
C/T, C/C	1073	73.09	173	76.44	1.20	0.87–1.67	0.27
** *Recessive* **							
T/T, C/T	1107	75.41	163	71.56		1.00	
C/C	361	24.59	63	28.44	1.19	0.87–1.62	0.29
** *Overdominant* **							
T/T, C/C	756	51.5	116	52		1.00	
C/T	712	48.5	110	48	1.01	0.76–1.33	0.96
**rs6314** ** *HTR2A Genotype* **							
C/C	1441	99.93	226	100		1.00	
C/T	2	0.07	0	0	0.00	0.00–NA	0.45
** *Allele* **							
C	2884	99.86	452	100			
T	2	0.14	0	0	NA	NA	0.44
** *Dominant* **							
A/A	375	25.7	64	28.3	1.00		0.4
C/A, C/C	1085	74.3	162	71.7	0.87	0.64–1.20	
** *Recessive* **							
A/A, C/A	1094	74.9	179	79.2	1.00		0.16
C/C	366	25.1	47	20.8	0.78	0.56–1.11	
** *Overdominant* **							
A/A, C/C	741	50.8	111	49.1	1.00		0.65
C/A	719	49.2	115	50.9	1.07	0.81–1.41	
**rs6296** ** *HTR1B* ** ** *Genotype* **							
C/C	348	25.18	63	28.44		1.00	
C/G	716	51.81	111	47.56	0.86	0.6–1.20	0.66
G/G	318	23.01	52	24	0.90	0.61–1.34	
** *Allele* **							
C	1412	51.09	237	52.22		1.00	
G	1352	48.91	215	47.78	1.06	0.87–1.29	0.59
** *Dominant* **							
C/C	348	25.18	63	28.44		1.00	
C/G, G/G	1034	74.82	163	71.56	0.87	0.64–1.19	0.39
** *Recessive* **							
C/C, C/G	1064	76.99	174	76		1.00	
G/G	318	23.01	52	24	1.00	0.72–1.40	0.99
** *Overdominant* **							
C/C, G/G	666	48.19	115	52.44		1.00	
C/G	716	51.81	111	47.56	0.90	0.68–1.19	0.45
**rs6265 *BDNF Genotype***							
G/G	432	29.61	65	28.44		1.00	
G/A	709	48.59	104	46.67	0.97	0.70–1.36	0.52
A/A	318	21.8	57	24.89	1.19	0.81–1.75	
** *Allele* **							
G	1573	53.91	234	51.78		1.00	
A	1345	46.09	218	48.22	1.09	0.89–1.33	0.40
** *Dominant* **							
G/G	432	29.61	65	28.44		1.00	
G/A, A/A	1027	70.39	161	71.56	1.04	0.76–1.42	0.79
** *Recessive* **							
G/G, G/A	1141	78.2	169	75.11		1.00	
A/A	318	21.8	57	24.89	1.21	0.87–1.67	0.26
** *Overdominant* **							
G/G, A/A	750	51.41	122	53.33		1.00	
G/A	709	48.59	104	46.67	0.90	0.68–1.19	0.47
**rs11030101** ** *BDNF* ** ** *Genotype* **							
A/A	691	47.3	135	61.82		1.00	
A/T	642	43.94	75	33.18	0.60	0.44–0.81	0.0004
T/T	128	8.76	11	5	0.44	0.23–0.84	
** *Allele* **							
A	2024	69.27	345	78.41		1.00	
T	898	30.73	97	21.59	0.63	0.50–0.80	0.0002
** *Dominant* **							
A/A	691	47.3	135	61.82		1.00	0.0004
A/T, T/T	770	52.7	86	38.18	0.57	0.43–0.76	
** *Recessive* **							
A/A, A/T	1333	91.24	210	95		1.00	0.043
T/T	128	8.76	11	5	0.55	0.29–1.03	
** *Overdominant* **							
A/A, T/T	819	56.06	146	66.82		1.00	0.0046
A/T	642	43.94	75	33.18	0.66	0.49–0.88	

These data represent N (%) by chi-square test, significant *p* value < 0.05, *TPH1*: Tryptophan hydroxylase 1, 2, *SLC6A4*, *HTR2A*: Serotonin receptor 2A gene, *HTR2A*: Serotonin receptor 2A gene, *HTR1B*: Serotonin receptor 1B gene, and *BDNF*: brain-derived neurotrophic factor gene.

## Data Availability

The original contributions presented in this study are included in the article. Further inquiries can be directed to the corresponding author.

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
