# Peer review of "Genetic Association Study of Brain-Derived Neurotrophic Factor (BDNF) and Serotonin-Related Gene Variants in Suicide Attempters with Major Depressive Disorder and Control Persons"

_genes, 2025, doi:10.3390/genes16111304_

Round 1
Reviewer 1 Report
Comments and Suggestions for Authors
Suicide attempts have been one of the most pressing public concerns in recent years. The etiology of suicidal behavior is understood to involve a complex interplay between genes and environmental factors. In this study, the authors investigate whether specific BDNF and serotonin-pathway gene polymorphisms are related to suicide attempts among individuals diagnosed with major depressive disorder (MDD). They genotyped a panel of candidate SNPs in suicide attempters recruited from an emergency department and in a larger control cohort. The analyses revealed a significant association between genes and suicide attempt status. Overall, the study is clearly structured and methodologically sound, presenting a coherent and well-supported link between genetic variation and suicidal behavior in MDD.
Comments:
Table 1. There appears to be a major data inconsistency: the mean age of the control group (~10 years) is far lower than that of the suicide attempters (~43 years). If accurate, this discrepancy weakens direct comparison because allele frequencies.
Table 3-1. Additional information is needed to better interpret the listed SNPs. Specifically, identify whether any variants show a higher frequency in the control group than in the suicide-attempt group, and discuss the potential biological relevance of such findings.
Table 3-5. Clarify whether the analyzed SNPs are located in coding or noncoding regions of their respective genes, and specify whether any of them result in amino acid substitutions or other functional changes.
Table 3-8. The table highlights genes with statistically significant associations, but it is unclear why these results are presented separately rather than integrated into Table 3-5. Consider merging the data or clearly explaining the rationale.
Author Response
attached fille..

Reviewer 2 Report
Comments and Suggestions for Authors
The manuscript addresses a relevant topic concerning the genetic association between BDNF and serotonin-related genes in patients with major depressive disorder and suicidal behavior.
Ethical approval and informed consent are clearly stated. The methodology is transparent, and data availability is adequate.
- I suggest performing statistical adjustments for age and sex, considering the major demographic imbalance between the study groups, and clarifying that the reported significances do not remain after Bonferroni correction.
- I recommend verifying and harmonizing the statistical values between the text and tables. Please carefully check all reported values (p-values, OR, 95% CI) and ensure consistent numeric formatting throughout. I also recommend using a dot as the decimal separator and limiting to two–three significant digits, in accordance with MDPI style (e.g., p = 0.004).
- I also suggest expanding the biological interpretation to highlight possible mechanisms through which BDNF rs11030101 and SLC6A4 rs3813034 may influence suicide risk.
- It would be useful to include additional clinical information regarding depression severity and treatments administered.
- I further propose discussing potential regional differences between the samples and performing minor linguistic editing for clarity.
- The discussion is overly descriptive and focuses mainly on prior studies without sufficiently integrating the biological meaning of the present findings. The authors should elaborate on mechanistic explanations and avoid overinterpreting nominally significant results.
- Tables are comprehensive but require clearer formatting and alignment.
The manuscript presents an original and clinically relevant investigation into the genetic association between BDNF and serotonin-related polymorphisms in patients with major depressive disorder who have attempted suicide. The topic is timely and fits well within the journal’s scope. The methodological approach is generally sound, and the identification of BDNF rs11030101 as a potential susceptibility locus adds novelty to the field.
However, there are several important limitations that need to be addressed before publication. The most critical issue is the pronounced demographic imbalance between the study and control groups (adult patients vs. adolescent controls), which may introduce significant confounding effects. Additionally, multiple testing correction nullifies most of the reported associations, and this should be clearly reflected in the discussion and conclusions. The presentation of statistical data requires careful verification, and the interpretation of biological mechanisms remains overly descriptive and should be strengthened.
Recommendation: Major Revision
Author Response
attached file..
